# Immune Phenotypes of Nasopharyngeal Cancer

**DOI:** 10.3390/cancers12113428

**Published:** 2020-11-18

**Authors:** Johan S. Nilsson, Aastha Sobti, Sabine Swoboda, Jonas S. Erjefält, Ola Forslund, Malin Lindstedt, Lennart Greiff

**Affiliations:** 1Department of ORL, Head & Neck Surgery, Skåne University Hospital, 221 85 Lund, Sweden; sabine.swoboda@skane.se (S.S.); lennart.greiff@skane.se (L.G.); 2Department of Clinical Sciences, Lund University, 221 85 Lund, Sweden; 3Department of Immunotechnology, Lund University, 223 81 Lund, Sweden; aastha.sobti@immun.lth.se (A.S.); malin.lindstedt@immun.lth.se (M.L.); 4Department of Experimental Medicine, Lund University, 221 84 Lund, Sweden; jonas.erjefalt@med.lu.se; 5Department of Microbiology, Lund University, 221 85 Lund, Sweden; ola.forslund@med.lu.se

**Keywords:** cancer immune phenotype, CD8, CD207, quantification, survival

## Abstract

**Simple Summary:**

As for many solid cancers, nasopharyngeal cancer (NPC) interacts with the immune system. In this retrospective study, immune features of NPC were explored and assessed against Epstein-Barr virus status, clinical stage, and survival. Specific immune phenotypes were identified based on presence and distribution of CD8+ T-cells: i.e., “inflamed”, “excluded”, and “deserted” NPC, which carried important prognostic information. Presence and distribution of CD207+ cells, likely representing antigen-presenting dendritic cells, were demonstrated, suggesting a potential for immune cell targeting. Gene expression revealed differences in immune profiles between NPC and control tissue as well as between subgroups of NPC based on CD8 expression (high vs. low). Taken together, the observations may be of relevance to prognostication of NPC as well as for explorations into the field of immunotherapy.

**Abstract:**

Nasopharyngeal cancer (NPC) features intralesional immune cells, but data are lacking on presence/distribution of T-cells and dendritic cells (DCs). Based on intralesional distribution of lymphocytes, a series of NPC biopsies (*n* = 48) were classified into “inflamed”, “excluded”, and “deserted” phenotypes. In addition, CD8+ T-cells and CD207+ DCs were quantified. The data were analyzed in relation to Epstein–Barr virus-encoded small RNA (EBER), Epstein-Barr virus (EBV) DNA, and survival. Separately, data on gene expression from a public database were analyzed. 61.7% of NPC lesions were “inflamed”, 29.8% were “excluded”, and 8.5% were “deserted”. While CD8+ cells were present in cancer cell areas and in surrounding stroma, CD207+ cells were observed largely in cancer cell areas. High CD8+ T-cell presence was associated with EBV+ disease, but no such pattern was observed for CD207+ DCs. There was a difference in disease-free survival in favor of “inflamed” over “excluded” NPC. Gene expression analysis revealed differences between NPC and control tissue (e.g., with regard to interferon activity) as well as between subgroups of NPC based on CD8 expression (high vs. low). In conclusion, NPC lesions are heterogeneous with regard to distribution of CD8+ T-cells and CD207+ DCs. NPC can be classified into immune phenotypes that carry prognostic information. CD207+ DCs may represent a target for immunotherapy with potential to facilitate the antigen cross-presentation necessary to execute cytotoxic T-lymphocyte responses.

## 1. Introduction

There is a need for new prognostic options and treatment principles for nasopharyngeal cancer (NPC), a malignancy frequently associated with Epstein–Barr virus (EBV), and measures targeting the immune system may offer such possibilities. However, this requires detailed knowledge about the cancer and its local microenvironment, particularly on the presence and distribution of intralesional immune cells and their targets and functions. In a recent study by Wang et al., the importance of the immune status of NPC was underscored: focusing on immune checkpoints PD-L1 and B7-H4 on tumor cells and PD-L1, B7-H3, B7-H4, IDO-1, VISTA, ICOS, and OX40 on intralesional immune cells, specific signatures were demonstrated to predict survival [1]. Interestingly, the association was particularly strong for patients with high levels of EBV DNA in plasma, suggesting the importance of EBV-antigen presence to the cancer and immune system interaction and indicating a need to monitor EBV status or even antigen levels in studies focusing on aspects of the immune system in NPC. Furthermore, in an even more recent study, Chen et al. demonstrated that specific gene signatures of macrophages, plasmacytoid dendritic cells (DCs), CLEC9A+ DCs, natural killer cells, and plasma cells were associated with improved progression-free survival [2].

With regard to interactions between the immune system and NPC, tissue-infiltrating lymphocytes (TILs) are of importance. The density/distribution of TILs has been investigated and identified as an independent positive prognostic factor [3]. Arguably, this lymphocyte population includes T-cells that, if appropriately instructed by DCs (e.g., via adjuvant targets such as C-lectin receptors), can facilitate antigen cross-presentation and can produce cytotoxic T-cell (CTL) responses [4,5]. Furthermore, the presence and distribution of lymphocytes, notably T-cells, as demonstrated for other cancers and indicated to predict, e.g., response to immunotherapy [6], may allow for classification of NPC into specific immune phenotypes. In addition to T-cell observations, following morphological findings suggesting a presence of DCs in NPC lesions [7,8,9,10,11,12], we recently reported on intralesional DC subsets and particularly highlighted CD1c+ myeloid cells expressing the C-lectin receptor CD207 [13]. However, taken together, data on the presence of T-cells, in particular CD8+ T-cells, and CD207+ DCs in NPC are scarce. Furthermore, it is unknown whether immune phenotypes exist in NPC and, if so, whether they influence survival. 

In this study, biopsy material from patients with NPC, of which we have previously reported data on intralesional levels of EBV DNA and survival [14], were revisited. In order to assess aspects of tumor heterogeneity, overall histology was examined focusing on cancer cells, cytokeratin, and Epstein–Barr virus-encoded small RNAs (EBERs). Based on the presence and distribution of lymphocytes (notably CD8+ cells), the material was classified into immune phenotypes. In addition, using a digital image technique, CD8+ and CD207+ cells were quantitatively assessed. The impact on survival of immune phenotypes as well as of CD8+ T-cells and CD207+ DCs was determined. The data were also analyzed in relation to intralesional EBV DNA. Finally, data on gene expression in the context of NPC and healthy control tissue were retrieved from a public database [15,16] and analyzed in silico focusing on CD8+ low and CD8+ high T-cell subgroups of NPC.

## 2. Results

### 2.1. Material Availability and Success Rate with Regard to Analyses

For all 48 patients, EBER slides were retrieved and new slides were produced for visualization of CD8+ and CD207+ cells, respectively. One individual was excluded from the analysis due to insufficient material for immunohistochemistry (i.e., the obtained tissues were deemed too small and of poor quality): this patient was incidentally also the only one lacking follow-up. For the remaining 47 patients, it was possible to perform all but three analyses. Two of these were related to a case of a heavily EBER-positive tumor with dense infiltrates of lymphocytes, making a distinction of nontumor-infiltrated surrounding stroma (CD8 and CD207 analyses) too unreliable. The third was an analysis of CD207 where areas of surrounding stroma were lacking.

### 2.2. Overall Histology and Cytokeratin/EBER Immunohistochemistry

In Figure 1, selected NPC lesions are presented focusing on presence and distribution of cancer cells, cytokeratin, and EBERs. With regard to cancer cells, a marked inter- and intraindividual heterogeneity was observed. Accordingly, some NPC lesions (or parts of lesions) featured multiple but aggregated areas of cancer cells with pushing borders, whereas others (or other parts) were characterized by cancer cells infiltrating the surrounding stroma. A heterogeneity was seen also for the presence and distribution of cytokeratin (an epithelial cell marker), in this context a marker of cancer cells.

### 2.3. Immune Phenotypes: “Inflamed”, “Excluded”, and “Deserted”

Immune phenotypes of the NPC lesions were estimated based on qualitative assessment of the presence and distribution of lymphocytes on slides stained for hematoxylin, cytokeratin, and CD8. Most lesions (61.7%) were of an immune “inflamed” phenotype (lymphocytes infiltrating cancer cell areas). Immune “excluded” (lymphocytes in the surrounding stroma but not infiltrating cancer cell areas) and “deserted” phenotypes (no lymphocytes in either cancer cell areas or the surrounding stroma) were less frequent: 29.8% and 8.5%, respectively. In Figure 2, selected NPC lesions are presented, demonstrating these three immune phenotypes.

### 2.4. CD8 and CD207: Quantification in Whole Biopsies

Following digital image dissection of *whole biopsies* (excluding artefacts, normal epithelium, germinal centers, and gland structures), areas eligible for analysis of CD8 and CD207, respectively, were 14.8 (6.5–28.7) mm^2^ and 12.3 (5.5–22.4) mm^2^. The frequencies of pixels representing CD8 and CD207, respectively, were 2.17% (0.73–3.18) and 0.15% (0.040–0.58). There was no correlation between the CD8 and CD207 ratios. The morphological distribution patterns of CD8 and CD207 are depicted in Figure 3, and the pixel frequencies are indicated in Figure 4a. Based on ratios of pixels representing CD8, there were differences between the immune phenotypes: “inflamed” and “excluded” (*p* = 0.034, higher frequency for “inflamed”), “inflamed” and “deserted” (*p* = 0.0020, higher frequency for “inflamed”), as well as “excluded” and “deserted” (*p* = 0.022, higher frequency for “excluded”). No such differences were observed for ratios of pixels representing CD207 (Figure 5a,b).

### 2.5. CD8 and CD207: Quantification in Areas of Cancer Cells and Surrounding Stroma

For quantification of CD8+ cells, defined areas were selected representing *surrounding stroma* and the frequency of pixels was 2.27% (0.91–4.94) (Figure 4b). For quantification of CD207+ cells, defined areas were selected representing *cancer cells* and *surrounding stroma*. The frequencies of pixels representing CD207 were 0.25% (0.040–1.39) for areas of cancer cells and 0.030% (0–0.065) for surrounding stroma, representing an 8-fold difference (*p* < 0.0001) (Figure 4c). There was no statistically significant correlation between ratios of CD8 or CD207 in defined areas (grouped by median values) and immune phenotypes. There was no correlation between the CD8 and CD207 ratios in the surrounding stroma.

### 2.6. Clinical Performance Based on Immune Phenotype and Presence of CD8 and CD207

The “deserted” phenotype comprised four NPC lesions that all were EBER-negative. Out of these, three featured spread disease at diagnosis (stage IVC) and one featured an advanced local lesion (stage IVA). There was no difference in disease-specific survival (DSS) between immune phenotypes, but a there was a statistically significant difference in disease-free survival (DFS) between the “inflamed” and “excluded” phenotype (*p* = 0.0090) (Figure 6), the latter presenting the poorest prognosis. Since three out of four cases of immune “deserted” phenotype presented as spread disease, this subset was not included in the DFS analysis. There was no difference in DSS or DFS between high and low quantitates of CD8 and CD207 (grouped by median levels).

### 2.7. Cancer Stage in Relation to Immune Phenotypes and to CD207 and CD8

In line with the findings that cases with a “deserted” phenotype were all stage IV disease at diagnosis (*n* = 4) and that cases of stage I disease (*n* = 4) were all of the “inflamed” phenotype, there was a difference between stage I–III and IV disease with regard to phenotypes (*p* = 0.042). No statistically significant associations were observed for other selected stage combinations or T-stage differences. There were no statistically significant differences between stage or stage combinations, and CD8+ or CD207+, either for whole biopsies or for selected regions.

### 2.8. EBV in Relation to Immune Phenotypes and to CD8 and CD207

The CD8 ratios for whole biopsies (Figure 7a) were higher for EBER-positive *cf*. -negative NPC lesions (*p* = 0.0065) and for EBV DNA-positive *cf*. -negative NPC lesions (*p* = 0.028). In contrast, there were no differences in CD207 ratios for whole biopsies (Figure 7b) and neither between CD207 ratios for either EBER or intralesional EBV DNA in selected areas (areas of cancer cells and areas of surrounding stroma) nor between CD8 ratios for EBER and intralesional EBV DNA in surrounding stroma. EBER-negative cases (*n* = 12) aggregated as “deserted” > “excluded” > “inflamed”: “Inflamed” vs. “excluded” (*p* = 0.0045), “inflamed” vs. “deserted” (*p* < 0.0001), and “excluded” vs. “deserted” (*p* = 0.043). For intralesional EBV DNA (present or not present), a difference was shown between “inflamed” vs. either “excluded” (*p* = 0.016, present associated with “inflamed”) or “deserted” (*p* < 0.0001, present associated with “inflamed”), while there was no such difference for “excluded” vs. “deserted”. A similar pattern was seen for quantitative EBV DNA data (Figure 8), where a DNA load difference was present between the “inflamed” and “deserted” phenotypes (*p* = 0.00034, higher load for “inflamed”).

### 2.9. Cell Type-Specific Gene Expression in NPC

Transcriptional data available from the gene expression omnibus (GEO) database, including mRNA profiles of 31 NPC samples and 10 control nasopharyngeal samples, were in silico immunoprofiled and further assessed based on CD8+ T cell-related transcripts. Utilization of the CD8+ T-cell signatures, based on cell profiling scores from Puram et al. and Newman et al. via CIBERSORTX [17,18], enabled a subdivision of the NPC samples into groups of high and low scores. The immune profiling scores displayed a significant increase in CD4+ T cells in controls *cf.* both CD8+ high and low NPC, higher fibroblasts score in CD8+ low NPC *cf*. control, and a significant increase in macrophages for CD8+ high NPC *cf.* controls (Figure 9a). On conducting a similar analysis of 22 immune cell populations [17], enhanced expression of the signatures of CD4+ memory activated T cells and M1 macrophages were evident in the CD8+ low and CD8+ high group *cf.* controls. Further, the relative intensity weights for the signature of Natural Killer (NK)-activated cells were increased in CD8+ high NPC *cf.* controls. In contrast, signatures for the naïve and memory B cells were significantly lower in both NPC groups *cf.* controls (Figure 9b). The DCs (resting and activated) could not be determined for NPC *cf*. control tissue. 

As the CD8A gene is the most representative functional gene associated with CD8+ T cells [19], further investigation for genes correlated to CD8A was conducted. The analysis revealed 25 positively and 1 negatively correlated gene to CD8A. The expression score of these genes with the 10 immune populations [18] showed a significant association with CD4+ T cells and mast cells. In addition, LAG3 was relatively correlated to the signatures of macrophages, B cells, and malignant cells. The expression levels of the CD244 gene, a cell surface receptor expressed on NK cells, T cells, and DCs [20,21], were associated with macrophages, DCs, and malignant cells. The only negative correlated gene, i.e., TALDO1, associated with fibroblasts as well as malignant cells and DCs (Figure 9c). Interestingly, for the 22 immune cell populations, all positively correlated genes exhibited significant association with gamma delta T cells and TALDO1 with the M2 macrophage population (Figure 9d). 

Gene expression data analysis showed that 48% (15/31) of the NPC cases displayed an interferon signature [22,23] in contrast to the remaining NPC samples and control tissue (Figure 9e). Out of these, 12 belonged to the CD8+ high group.

## 3. Discussion

In this study, primary biopsies from patients with NPC were analyzed. A marked inter- and intraindividual variation was observed with regard to growth pattern of cancer cells as well as to presentation of cytokeratin, EBERs, and immune cells. Based on presence and distribution of lymphocytes as established for other cancers [6], the lesions were classified into specific immune subsets, which carried prognostic information: DFS was better for the “inflamed” than for the “excluded” phenotype, while the “deserted” phenotype, arguably with the worst prognosis, was not eligible for analysis due to a majority of cases presenting with spread disease. CD8+ cells were present in areas of cancer cells and in the surrounding stroma, whereas CD207+ cells were observed largely in areas of cancer cells. The ratio of CD8+ cells were higher for EBV-positive *cf*. EBV-negative NPC. In contrast, no such differences were observed for CD207. Gene expression analysis revealed differences between NPC and control tissue (e.g., with regard to interferon activity) as well as between subgroups of NPC based on CD8 expression (high vs. low). Taken together, the observations may be of relevance to prognostication of NPC as well as for explorations into the field of immunotherapy.

The notion that head and neck cancer lesions, including NPC, are heterogeneous in nature (e.g., Wang et al. [3]), was confirmed by this study. With regard to patterns of cancer cell growth, this was evident for inter- as well as intraindividual comparisons. Inferentially, a similar heterogeneity was observed for cytokeratin (which in this context may be viewed as a cancer cell marker), which is typically expressed by NPC cells [24]. These observations, in combination with occasional findings of diffuse EBER patterns, indirectly suggest the possibility that intralesional cancer-associated antigen-levels, as previously suggested through analysis of EBV-DNA [14], and immune responses may also vary considerably. In this study, such a marked variability was observed for the presence and distribution of CD8+ T-cells and CD207+ DCs. Taken together, our observations suggest that NPC heterogeneity, including immune cell aspects, must be taken into account when, e.g., prognostic information is explored and candidate treatment targets are selected for this condition. Arguably, even individual information may be of importance to future immunological treatment possibilities. The use of digital image “microdissection” and quantitative digital analysis in this study represents efforts in that direction. However, a limitation of the technique is that it does not allow for quantitation of CD8+ cells in cancer nodules due to difficulties in discriminating between areas of cancer cells infiltrated by lymphocytes and the lymphocyte-rich surrounding stroma.

In this study, as previously suggested for other cancers [6,25], the presence and distribution of lymphocytes (notably CD8+ T-cells) was utilized to classify NPC into specific immune phenotypes. The majority of NPC lesions were of an immune “inflamed” nature (61.7%), with lymphocytes infiltrating areas of cancer cells, while the minority was either immune “excluded” (29.8%) or “deserted” (8.5%), a classification that was verified by quantitative analysis of CD8+ T-cells. Of the immune phenotypes, the “deserted” subset was associated with the poorest prognosis. A high frequency of spread disease at diagnosis excluded this subset from an analysis of DFS. However, a clear difference in DFS was observed between the “inflamed” and the “excluded” phenotypes (in favor of the former). DSS showed a similar pattern, but the differences failed to reach statistical significance. We suggest that immune phenotype information should be evaluated further as a prognostic marker for NPC and in the context of treatment selection, for example, whether to combine first-line treatment (i.e., radiotherapy) with chemotherapy for the “deserted phenotype” even for low-stage disease or to offer checkpoint inhibitor immunotherapy to patients with NPC manifesting the “inflamed” phenotype, since observations in other types of cancers suggest that this phenotype is associated with a positive treatment response to such interventions [26,27]. Our observations extend previous work in the field: for example, the impact of TILs in NPC and their potential as prognostic markers [3]. However, a limitation of our study is its sample size, which does not allow for a comprehensive analysis of survival.

The DC is specialized in antigen presentation and for a successful adaptive immune response to occur; either by the immune system through its own capacity or facilitated by “vaccinations”, this cell type needs to be present and active. Through this study, as CD207 is considered a selective DC marker [28,29], our previous observation that NPC lesions feature CD207+ DCs was confirmed (likely reflecting CD1c+ myeloid cells) [13]. Importantly, our present observations added the information that these cells were largely distributed within the areas of cancer cells, i.e., the frequency of CD207+ cells was 8-fold greater in this compartment *cf*. the surrounding stroma. However, while CD207+ DCs were present in close relation to cancer cells, this was apparently not enough to induce a meaningful immunological response targeting cancer antigens, which would prevent NPC from occurring or to be killed off. We suggest that the DC presence in NPC represents treatment possibilities and that the C-lectin receptor CD207 and potentially other pattern recognition receptors known to facilitate cross-presentation of antigen [4,5] may be adjuvant targets. However, additional DC subsets must also be considered, and any target and its function must be considered in relation to the overall milieu of the NPC lesion, including aspects that exert immunosuppressive actions that may prevent antigen presentation. Furthermore, whether CD207 has a role in EBV-specific T-cell responses remains to be shown.

The interplay between the immune system and cancer cells in NPC is not sufficient to eradicate the disease despite the fact that antigens are present and that key immune cells such as DCs and T-cells are available. More information is needed, and the present gene expression analysis, performed on available transcriptional data from EBV-positive NPC and control tissue [17,18], highlighted some immune features associated with NPC. High expressions of signatures related to interferon activity, M1 macrophages, and CD4+ memory activated T-cells were observed for NPC (*cf.* controls), albeit with marked heterogeneity between NPC samples. When comparing high vs. low CD8 vs. controls, high fibroblasts scores for CD8 low NPC were revealed. Furthermore, a high activated Natural Killer (NK) cell profile for CD8 high NPC was observed. (DCs could not be determined for NPC *cf*. control tissue.) Taken together, our observations suggest that there are immunological subsets of NPC and distinctions between NPC rich in CD8 (likely representing an “inflamed” phenotype) and low in CD8 (“excluded” and particularly “deserted” phenotypes). However, the gene expression data are difficult to put into a greater context given the lack of synchronous morphological assessment and clinical data, reflecting a general problem with gene expression data extracted from public databases.

In a previous study, we explored the association between intralesional EBV DNA and survival [14]. When the material was split at a level of 70 copies of EBV DNA per cell, higher levels predicted a greater DFS. In agreement, in this study, when intralesional EBV DNA load was assessed in relation to immune phenotypes, high loads were associated with the “inflamed” phenotype and low loads with the “deserted” phenotype. For reasons yet to be defined, the associations to immune phenotypes appeared stronger when focusing on EBER than EBV DNA. Similarly, the association between CD8 ratios and “EBV status” was stronger for EBER than EBV DNA. Taken together, our observations confirm that overall immune features of NPC, notably the presence of CD8+ T-cells, depend on the presence of EBV. We suggest that analyses of EBV should be included whenever immune features of NPC are examined, and that intralesional EBV DNA should be complemented by analysis of EBER.

## 4. Materials and Methods 

### 4.1. Study Design and Patients

The study was of a retrospective design and involved an analysis of formalin-fixated paraffin-embedded (FFPE) primary tumor tissue from a well-defined population of 48 patients with NPC diagnosed between 2001 and 2015. Data from this material have been reported previously focusing on intralesional EBV DNA and survival [14]. Through immunohistochemistry and based on lymphocyte presence and distribution, the tumors were classified into immune phenotypes. Furthermore, CD8+ and CD207+ cells were assessed using a quantitative digital image technique. The data were analyzed in relation to EBER, intralesional EBV DNA, clinical stage, and survival. Approval was granted by the Ethics Committee at Lund University (ref. no. 2014/117). In accordance with the approval, informed consent was not required, but the study was advertised in printed media with a possibility to opt out. Separately, gene expression data in NPC and control tissue were retrieved from a public database and analyzed in silico.

### 4.2. Clinical Characteristics

Patient characteristics have been previously reported [14]. Briefly, out of the 48 available NPC patients, one was lost to follow-up, and the median follow-up of the remaining 47 patients was 6.4 years; 31% of the patients were diagnosed with T1 lesions, 85% were N-positive, and 19% featured distant disease at the time of diagnosis (UICC’s TNM classification system, 7th version). The 5- and 7-year overall survival (OS) was 75% and 65%, respectively. At histopathological examination, 75% of the tumors were EBER-positive non-keratinizing cancers. Eighty-three percent of the patients presented EBV DNA-positive lesions. Clinical data on the material retrieved for gene expression analysis in silico was restricted to diagnosis, EBER status (all were EBER positive), and stage (UICC’s TNM classification system, 6th version). In the latter material, comprising data on 31 NPC patients, 21 featured stage I and II disease, 10 stage III disease, and none stage IV disease.

### 4.3. Immunohistochemistry

A modified double-staining immunohistochemistry protocol (EnVision Doublestain, Dako/Agilent, Glostrup, Denmark) was applied in order to simultaneously identify cytokeratin-positive cancer cells together with CD207+ DCs or CD8+ T cells. Briefly, 4-µm rehydrated sections were subjected to double-staining immunohistochemistry in the automated Autostainer IHC-robot (Dako/Agilent) using the EnVision Doublestain System kit K5361 (Dako/Agilent). Prior to immunohistochemistry, an antigen retrieval procedure was performed in a PT-link machine for heat-induced epitope retrieval (HIER) using low pH (pH 6) retrieval buffer. Endogenous peroxidase activity was blocked with H_2_O_2_. Sections were then incubated with an anti-CD207 (clone 12D6, Novocastra/Leica, Newcastle, UK; dilution 1:300) or anti-CD8 (clone C8/144B, Dako/Agilent, Glostrup, Denmark; dilution 1:400) primary antibody for 1 h at room temperature. After subsequent incubation with polymer/HRP-linked secondary antibodies for 30 min, the immunoreactivity was visualized using diaminobenzidine 3.3 (DAB) HRP chromogen (resulting in a brown opaque staining). Next, sections were incubated with Dako Double Stain Blocking Reagent (Dako/Agilent) to prevent additional binding of secondary antibodies to the first primary antibody. Sections were then incubated with an anti-cytokeratin primary antibody (clone CK AE1/AE3, Novocastra/Leica, dilution 1:300). This second-round immunoreactivity was also visualized with a polymer/HRP-linked secondary antibody (Dako/Agilent, dilution 1:300), but the visualization was performed with Vina Green HRP chromogen (Biocare Medical, cat BRR807A, Pacheco, CA, USA). Finally, sections were counterstained with Mayer’s hematoxylin, air-dried, and mounted with Pertex (Histolab Products, Gothenburg, Sweden). In addition to the protocols above, previously stained slides for EBER evaluation (EBER-ISH for EBER1 and EBER2) were retrieved [14]. All slides were digitalized using the automated Scanscope XT digital slide scanner (Aperio Technologies, Vista, CA, USA). Evaluation and quantification were performed using the Aperio Imagescope Software (Aperio Technologies).

### 4.4. Immunoprofiling

Based on overall lymphocyte presence, and with special regards to CD8+ T-cell distribution (i.e., the assessment was performed on hematoxylin-slides stained for CD8), the lesions were classified according to immune phenotypes defined as “inflamed” (lymphocytes infiltrating cancer cell areas), “excluded” (lymphocytes in the surrounding stroma with no, or exceedingly few, infiltrating lymphocytes within cancer cell areas), and “deserted” (no or exceedingly few lymphocytes both in cancer cell areas and the surrounding stroma). In cases of highly heterogeneous lesions, the dominating pattern was chosen for the classification. This assessment was performed temporarily blinded to clinical data (J.S.N) and verified totally blinded by a second observer (S.S.): EBER slides were not used as support in the analysis.

### 4.5. Quantification of CD8 and CD207 Immunoreactivity

Slides stained for CD8 and CD207 were reviewed in a standardized manner utilizing cytokeratin (all slides were stained for cytokeratin) with EBER slides as support. Through digital image dissection, nonrelevant regions were excluded. These comprised physical artefacts (e.g., occasionally folded sections), normal epithelium, germinal centers, and gland structures. The dissections were performed on consecutive slides for CD8 and CD207, enabling a consequent and representative exclusion strategy. The Positive Pixel 9-algorithm (Aperio Technologies) was used to automatically segment out tissue background area as well as positive staining (i.e., brown DAB chromogen) by color threshold values, and then the ratio of the total analyzed tissue area with staining-positive pixels was calculated. Accordingly, this ratio represented the quantities of CD8 and CD207 in the target area. Thresholds to define staining (chromogen) positivity were the same for CD8 and CD207.

In addition, areas (minimum 0.1 mm^2^ after exclusion of nonrelevant regions as defined above) were selected to represent areas of cancer cells and areas of surrounding stroma (in direct proximity to cancer cells). CD207 positivity was quantitated for both locations. However, CD8 positivity was quantitated for the surrounding stroma only. The latter was due to a very high lymphocyte presence, which in combination with a variable degree of lymphocyte infiltration, rendered it impossible to quantitate CD8 within cancer cell areas in an objective and robust manner. When the presence of the marker (CD8 and DC207) was different between various areas of the tumor, the area with more pronounced presence was chosen. The selected areas were assessed for each marker following the same digital image quantification process as described above. All quantification procedures were performed blinded to the clinical data.

### 4.6. Gene Expression Data Analysis

Normalized mRNA data, generated with the Affymetrix Human Genome U133 Plus 2.0 Array, were obtained from the GEO database (https://www.ncbi.nlm.nih.gov/geo/: accession number GSE12452) [15,16]. The analyses were performed using Qlucore Omics Explorer 3.6 (Qlucore, Lund, Sweden). Data were divided into two subsets: control tissue (*n* = 10), comprising 4 cases of nasopharyngeal tissue adjacent to but separated from NPC lesions and 6 cases of nasopharyngeal tissue from non-NPC subjects, and NPC tissue (*n* = 31, all EBER-positive). 

Utilizing CIBERSORTX [30], the gene expression data was immunoprofiled using signatures identified in two independent studies: one including the overall distribution of 10 cell populations using data from single-cell analysis of head and neck cancers [18] and the other defining 22 general tumor-associated immune cell populations [17]. The scoring from both studies for CD8+ T-cell signatures were used to divide the NPC samples into CD8+ high and low. Also, the material was analyzed for CD8A-correlated genes and expression levels based on an interferon signature (39 genes). Heat map clustering was performed to understand the relationship between CD8A-correlated genes and individual cell populations [17,18].

### 4.7. Statistics

For the principal data set subjected to digital imaging, statistical analyses were performed using SPSS version 25 (IBM, Armonk, NY). Data were presented as medians with interquartile ranges (IQR). For comparisons of CD8 and CD207 ratios, respectively, between immune phenotypes of NPC and between tumor stages, an analysis of variance (the Kruskal–Wallis test) was followed by the Mann Whitney U-test. For comparisons of EBV DNA load between immune phenotypes, the Kruskal–Wallis test was followed by the Mann Whitney U-test. χ^2^-tests were performed to explore associations between CD8 and CD207 (grouped by median values) and EBER (present or not), EBV DNA (present or not), immune phenotypes, and stage. Correlations between CD8 and CD207 ratios and between these ratios and EBV-DNA were explored using the Spearman test. Comparisons of ratios for CD207 between areas of cancer cells and surrounding stroma was explored using the Wilcoxon Signed Ranks test. Survival, i.e., DSS and DFS in relation to CD8 and CD207, respectively, as well as to immune phenotypes, was described using Kaplan–Meyer curves, and significance levels were determined by the log-rank test. A Cox regression analysis was not performed due to the restricted sample size. The explicit *p*-value was provided when the *p*-value was <0.05 but >0.0001, whereas lower values were provided as <0.0001.

CD8A-correlated gene markers and the interferon signatures were examined using the limma multigroup analysis (edgeR) [31]. A Pearson correlation of r > 0.7 and adjusted *p*-values less than 0.01 were considered statistically significant. For gene expression data analyzed in silico, statistical analyses were performed using GraphPad version 8.4.3 (GraphPad Software, La Jolla, CA, USA). The intensity fractions obtained were analyzed for differences in the immune cell populations using the two-way ANOVA, Tukey method.

## 5. Conclusions

In conclusion, NPC lesions are heterogeneous with regard to the presence and distribution of cancer cells as well as immune cells (CD8+ T-cells and CD207+ DCs). In addition, there are immune-related differences between subgroups of NPC based on CD8 expression (high vs. low). Arguably, this nature, in addition to a variable presence of EBV-associated antigen, must be taken into account when prognostic information is examined and candidate treatment targets are explored. With regard to distribution of lymphocytes (notably CD8+ T-cells), NPC may be classified into “inflamed”, “excluded”, and “deserted” immune phenotypes, and these subsets carry prognostic information that is easily accessible and that can be linked to EBV status. CD207+ DCs, likely representing the myeloid CD1c+ subtype, are present in NPC lesions. Intralesional DCs may represent a possibility for immunotherapy, and CD207 may be a specific target based on its ability to facilitate a cross-presentation of antigens necessary to produce antigen-specific cytotoxic T-cell responses.

## Figures and Tables

**Figure 1 cancers-12-03428-f001:**
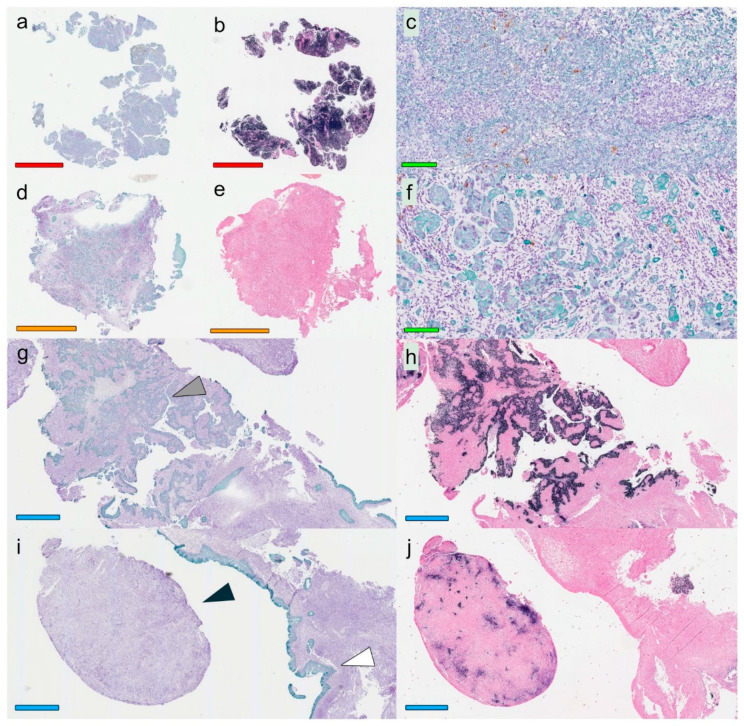
Overall histological view for three selected patients: *Patient 1* with EBER-positive NPC (**a**–**c**), *Patient 2* with EBER-negative NPC (**d**–**f**), and *Patient 3* with EBER-positive NPC (**g**–**j**). A marked heterogeneity was seen between patients (**all panels**) and within a single sample (**g**–**j**), reflected as varying expression of both CK and EBER. The stainings used were Mayer’s hematoxylin in combination with CK (green) and CD207 (brown) (**a**,**c**,**d**,**f**,**g**,**i**) and Red Counterstain II with EBER (black) (**b**,**e**,**h**,**j**). Colored horizontal bars indicate size: red = 3 mm, green = 100 µm, orange = 2 mm, and blue = 900 µm. Arrowheads (**g**,**i**) denote cancer cells expressing CK and EBER (gray), cancer cells with loss of CK (black), and normal CK-staining of epithelium as comparison (white). Abbreviations: EBER = Epstein–Barr virus encoded small RNAs, NPC = nasopharyngeal cancer, and CK = cytokeratin.

**Figure 2 cancers-12-03428-f002:**
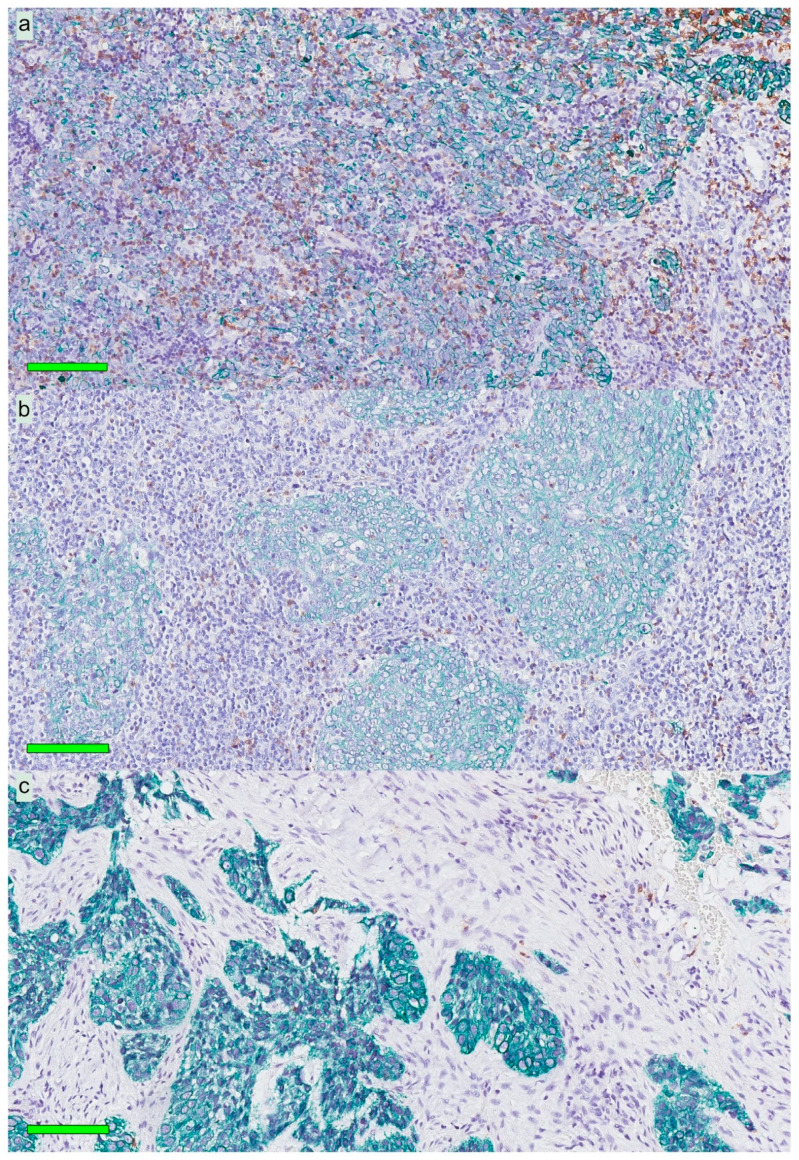
Immune phenotypes for three patients with NPC: Mayer’s hematoxylin in combination with CK (green) and CD8 (brown). All CK expressions denote cancer cells. The *top panel* (**a**) shows an “inflamed” tumor rich in infiltrating lymphocytes. The *middle panel* (**b**) indicates an “excluded” tumor with lymphocytes surrounding areas of cancer cells. The *bottom panel* (**c**) demonstrates a “deserted” tumor with no or very few lymphocytes in areas of cancer cells and the surrounding stroma. Green horizontal bars indicate 100 µm. Abbreviations: NPC = nasopharyngeal cancer and CK = cytokeratin.

**Figure 3 cancers-12-03428-f003:**
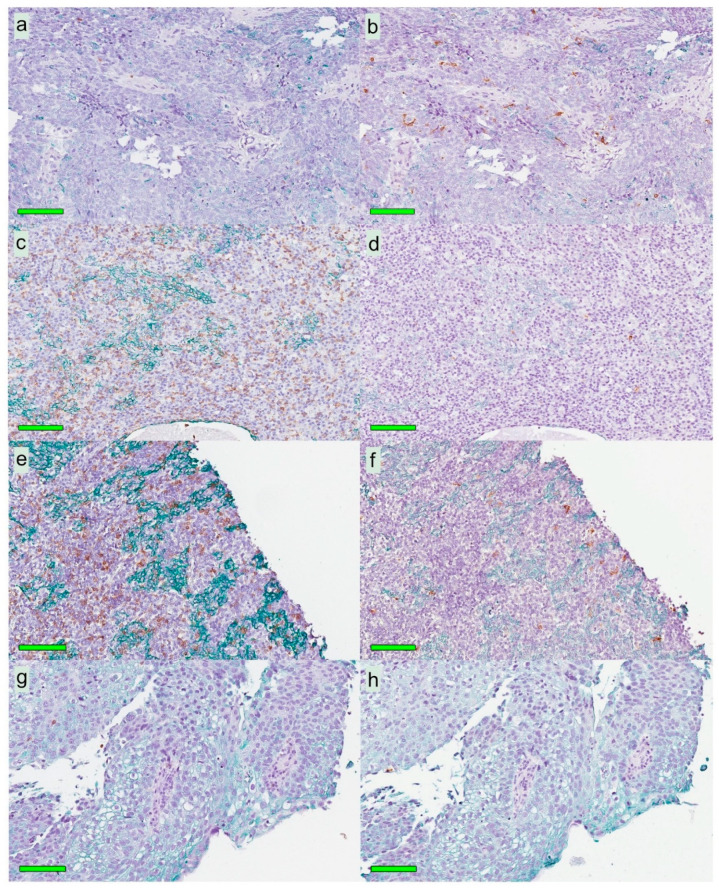
Different patterns of CD8+ and CD207+ cell distribution in four selected patients: consecutive tissue sections with Mayer’s hematoxylin in combination with CK (green) and either CD8 or CD207 (brown). Left panels indicate the distribution of CD8+ cells, and right panels indicate CD207+ cells on the following section. The patterns shown are (**a**,**b**) low in CD8 and high in CD207, (**c**,**d**) high in CD8 and low in CD207, (**e**,**f**) high in both CD8 and CD207, and (**g**,**h**) low in both CD8 and CD207. CD8 expression was in general higher than CD207 expression. Green horizontal size-bars indicate 100 µm. Abbreviations: NPC = nasopharyngeal cancer and CK = cytokeratin.

**Figure 4 cancers-12-03428-f004:**
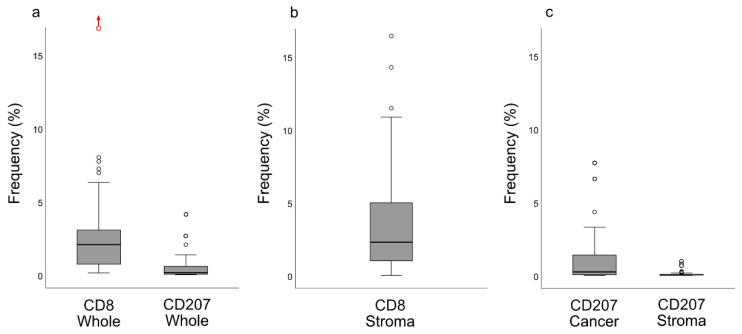
CD8 and CD207 frequency (%) presented as boxplots (median and IQR with whiskers denoting 1.5 IQR and outliers as circles) for the whole biopsies (**a**) as well as for selected areas: (**b**) CD8 and (**c**) CD207. There was a difference between CD207 ratios between areas of cancer cells and the surrounding stroma (*p* < 0.0001). In (**a**), an extreme outlier (frequency: 24%), deemed an accurate observation, is indicated with a red circle and an upwards pointing red arrow. Abbreviation: IQR = interquartile range.

**Figure 5 cancers-12-03428-f005:**
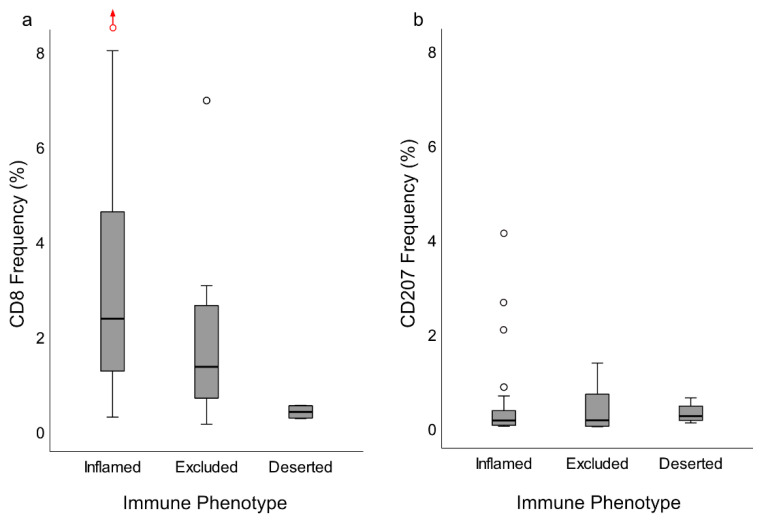
CD8 and CD207 frequency (%) of whole biopsies in relation to immune phenotypes presented as boxplots (median and IQR, with whiskers denoting 1.5 IQR and outliers as circles). (**a**) CD8 ratios differed between immune phenotypes: “inflamed” and “excluded” (*p* = 0.034, higher frequency for “inflamed”), “inflamed” and “deserted” (*p* = 0.0020, higher frequency for “inflamed”), and “excluded” and “deserted” (*p* = 0.022, higher frequency for “excluded”). (**b**) There was no difference in CD207 ratios between immune phenotypes. In (**a**), an extreme outlier (frequency: 24%), deemed an accurate observation, is indicated with a red circle and an upwards pointing red arrow. Abbreviation: IQR = Interquartile range.

**Figure 6 cancers-12-03428-f006:**
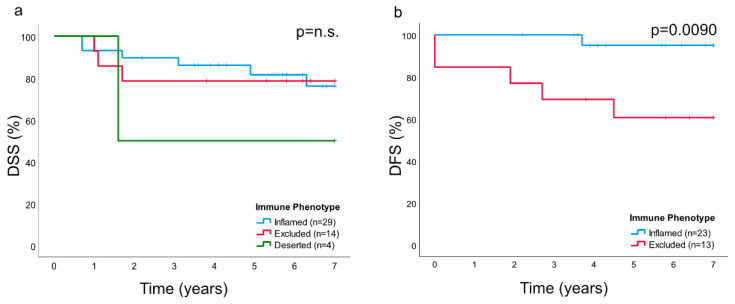
Kaplan–Meier estimates of (**a**) DSS and (**b**) DFS for NPC based on immune phenotypes indicated a better prognosis in terms of DFS for the “inflamed” subtype compared to the “excluded” (*p* = 0.0090). Since three out of four patients with immune “deserted” phenotype presented with spread disease, this subset was not included in the DFS analysis. No differences were observed in the DSS analysis. Abbreviation: DSS = disease-specific survival and DFS = disease-free survival.

**Figure 7 cancers-12-03428-f007:**
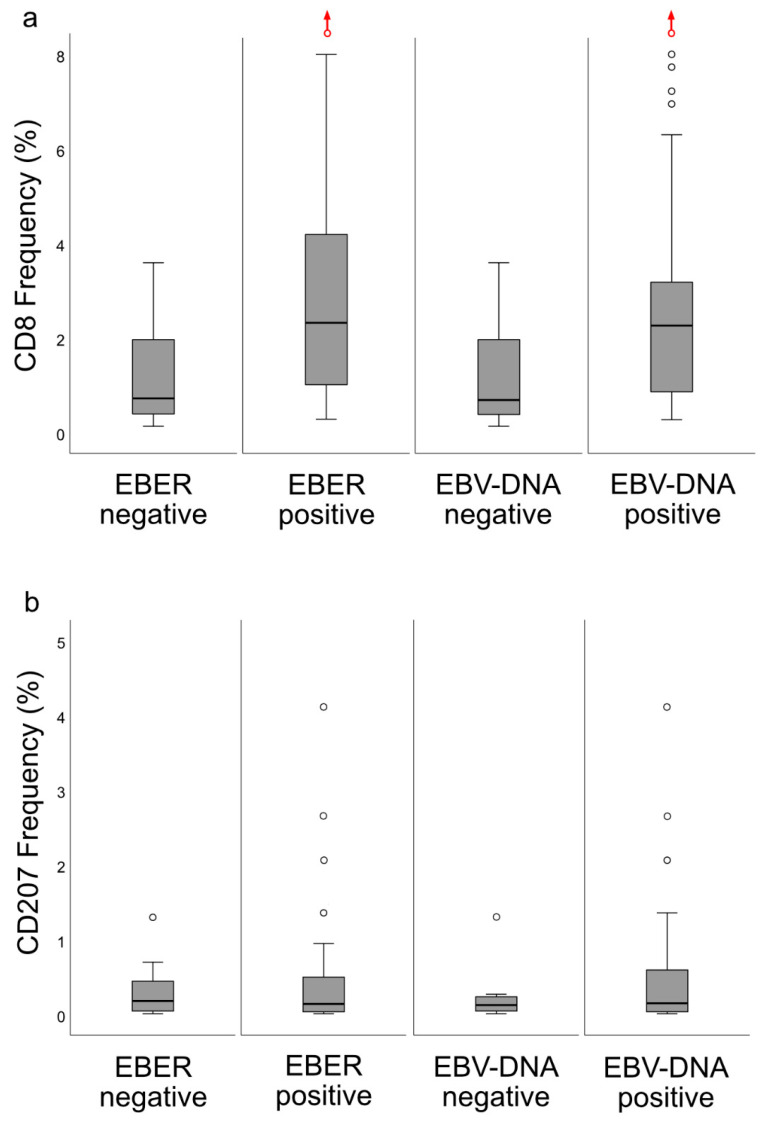
Frequency (%) for CD8 and CD207 (whole biopsies) grouped according to presence or not of intralesional EBER and EBV DNA, respectively, and presented as boxplots (median and IQR with whiskers denoting 1.5 IQR and outliers as circles): (**a**) CD8 ratios were higher for EBER-positive lesions (*p* = 0.0065) and for EBV DNA-positive lesions (*p* = 0.028); (**b**) in contrast, there were no such differences for CD207 ratios. An extreme outlier (frequency: 24%), deemed an accurate finding, is indicated with a red circle and an upwards-pointing red arrows. Abbreviations: EBER = Epstein–Barr virus-encoded small RNA, EBV = Epstein–Barr virus, and IQR = interquartile range.

**Figure 8 cancers-12-03428-f008:**
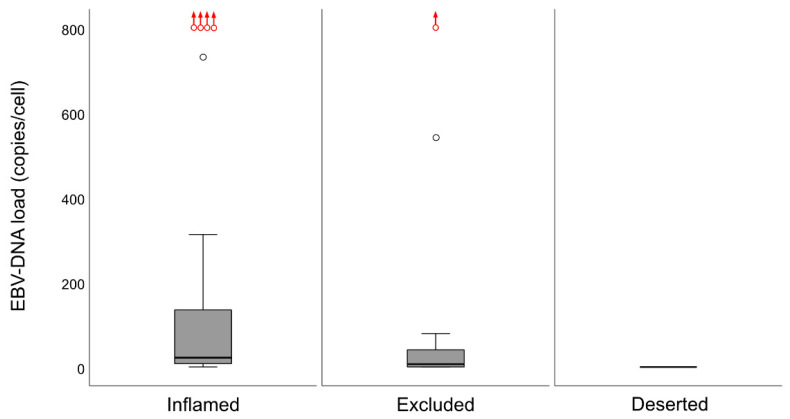
Differences in intralesional EBV DNA load (copies/cell) for immune phenotypes are shown (medians and IQR with whiskers denoting 1.5 IQR and outliers as circles). A marked difference in DNA load was present between the “inflamed” and “deserted” phenotypes (*p* = 0.00034). The differences between “inflamed” and “excluded” and between “excluded” and “deserted” were not significant, though a trend was seen (*p* = 0.055 and *p* = 0.079, respectively). Higher load outliers (range 1237–94617 copies/cell) are indicated with red circles and upwards-pointing red arrows. Abbreviation: EBV = Epstein–Barr virus.

**Figure 9 cancers-12-03428-f009:**
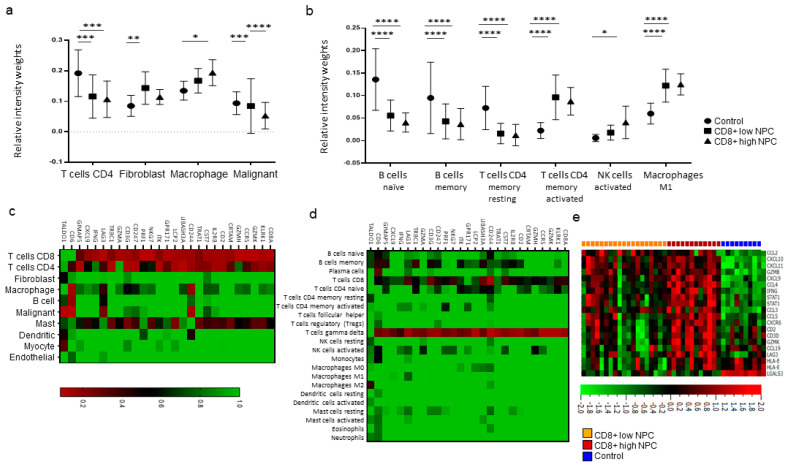
Immune cell profiling via gene expression for NPC (*n* = 31) and control tissue (*n* = 10) in the GEO dataset: (**a**,**b**) relative immune cell population distribution by CIBERSORTX in CD8+ high and low NPC compared to control tissue, using signatures from Puram et al. and Newman et al., respectively; (**c**,**d**) heat maps showing association of CD8A-correlated genes for 10 cell populations (Puram et al.) and 22 immune cell populations (Newman et al.); and (**e**) interferon gene signature in NPC *cf.* control, marked as orange, red, and blue for CD8+ low NPC, CD8+ high NPC, and control, respectively. The *p*-values in (**a**,**b**) are depicted as * < 0.01, ** < 0.005, *** < 0.001, and **** < 0.0001. Abbreviations: NPC = nasopharyngeal cancer and GEO = gene expression omnibus.

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
