# Peer review of "Immune Phenotypes of Nasopharyngeal Cancer"

_cancers, 2020, doi:10.3390/cancers12113428_

Round 1
Reviewer 1 Report
Thanks to the authors for making the changes requested following the first round of review. I am completely satisfied that these changes have elevated the paper to a status worthy of publication in Cancers journal.
Author Response
Thanks for your help regarding this paper.
Reviewer 2 Report
All of 48 patients’ numbers were too few, it may be difficult to collect the quantity, but at least 200 is recommended.
Author Response
In response to the question (Q3), originally put forward by Reviewer 2, now reiterated as:
“All of 48 patients’ numbers were too few, it may be difficult to collect the quantity, but at least 200 is recommended”,
We originally responded:
“Indeed, NPC is rare in Caucasian catchment areas, and the present data was collected over a 14-year period (as originally stated in Materials and Methods). Although we agree that a larger cohort would be necessary for certain analyses (e.g. Q4 below), the sample size was large enough to indicate statistically significant outcomes for key parameters in our analysis (acknowledged by Reviewer 1). However, in the revised version of the manuscript, we have now included a passage on limitation of our study in this context (page 12 lines 32-34). (Please, note that we cannot collect a number of 200 in our life time.)”
As per your instruction, we wish to specify our response with regards to which conclusions that can be drawn from the material despite its sample-size.
1 Clinical performance based on immune phenotypes:
Text in the ms: “There was no difference in disease-specific survival (DSS) between immune phenotypes, but a statistically significant difference in disease-free survival (DFS) between the “inflamed” and “excluded” phenotype (p=0.0090) (Figure 6), the latter presenting the poorest prognosis. Since three out of four cases of immune “deserted” phenotype presented as spread disease, this subset was not included in the DFS analysis.”
We fully agree with Reviewer 2 (our revised manuscript page 12 lines 32-34 was prompted by the original comment) that access to a greater number of patients might have enabled a more comprehensive analysis of survival (see §2 above for suggested change). However, our study is sufficiently powered to detect differences in disease-free survival. Furthermore, our study comprises many additional parts, for most of which the response by Reviewer 2 is irrelevant because a) these parts were purely descriptive (2-3, below), b) statistically significant differences did indeed emerge (4-6 below), or c) they were based on another study-material (7 below).
2 An overall non-comparative assessment of cytokeratin and EBER immunohistochemistry:
This part was purely descriptive and by design not subjected to comparative analysis. (The comment by Reviewer 2 may not be relevant in this context.)
3 A non-comparative classification of immune phenotypes: “inflamed”, “excluded” and “deserted”:
This part was also purely descriptive and by design not subjected to comparative analysis. (The comment by Reviewer 2 may not be relevant in this context.)
4 CD8 and CD207: quantification in whole biopsies:
Text in the ms: “Based on ratios of pixels representing CD8, there were differences between the immune phenotypes: “inflamed” and “excluded” (p=0.034, higher frequency for “inflamed”), “inflamed” and “deserted” (p=0.0020, higher frequency for “inflamed”), as well as “excluded” and “deserted” (p=0.022, higher frequency for “excluded”). No such differences were observed for ratios of pixels representing CD207.”
This part was comparative and statistically significant differences between immune phenotypes were observed for CD8+ T-cells. (As for CD207 cells, no discernable differences emerged at all.) (The data may not be questioned based on sample-size.)
5 CD8 and CD207: quantification in areas of cancer cells and surrounding stroma:
Text in the ms: “For quantification of CD8+ cells, defined areas were selected representing surrounding stroma and the frequency of pixels was 2.27% (0.91-4.94) (Figure 4b). For quantification of CD207+ cells, defined areas were selected representing cancer cells and surrounding stroma. The frequency of pixels representing CD207 was 0.25% (0.040-1.39) for areas of cancer cells and 0.030% (0-0.065) for surrounding stroma, representing an 8-fold difference (p<0.0001) (Figure 4c). There was no statistically significant correlation between ratios of CD8 or CD207 in defined areas (grouped by median values), and immune phenotypes. There was no correlation between the CD8 and CD207 ratios in the surrounding stroma.”
This part, for which digital imaging again was employed, was comparative and a statistically significant difference between areas of cancer cells and surrounding stroma was observed for CD207+ cells. (The data may not be questioned based on sample-size.) CD8+ T-cells was not compared betweeen the two sites for reasons discussed in the manuscript: this circumstance is not relevant in context of the comment of Reviewer 2.
6 EBV in relation to immune phenotypes and to CD8 and CD207
Text in the ms: “The CD8 ratios for whole biopsies (Figure 7a) were higher for EBER positive cf. negative NPC lesions (p=0.0065) and for EBV-DNA positive cf. negative NPC lesions (p=0.028). In contrast, there were no differences in CD207 ratios for whole biopsies (Figure 7b) and neither between CD207 ratios for either EBER or intralesional EBV-DNA in selected areas (areas of cancer cells and areas of surrounding stroma), and between CD8 ratios for EBER and intralesional EBV-DNA in surrounding stroma. EBER negative cases (n=12) aggregated as “deserted” > “excluded” > “inflamed”: “Inflamed” vs. “excluded” (p=0.0045), “inflamed” vs. “deserted” (p<0.0001), and “excluded” vs. “deserted” (p=0.043). For intralesional EBV-DNA (present or not present) a difference was shown between “inflamed” vs. either “excluded” (p=0.016, present associated with “inflamed”) or “deserted” (p<0.0001, present associated with “inflamed”), while there was no such difference for “excluded” vs. “deserted”. A similar pattern was seen for quantitative EBV-DNA data (Figure 8), where a DNA load difference was present between the “inflamed” and “deserted” phenotypes (p=0.00034, higher load for “inflamed”).”
This part, for which quantitative DNA assessments and digital imaging again was employed, was comparative and statistically significant differences emerged throughout as described above. (The data may not be questioned based on sample-size.)
7 Cell type-specific gene expression in NPC
Text in the ms: Figure 9. Immune cell profiling via gene expression for NPC (n=31) and control tissue (n=10) in the GEO dataset. (a-b) Relative immune cell population distribution by CIBERSORTX in CD8+ high and low NPC compared to control tissue, using signatures from Puram et al. and Newman et al., respectively. (c-d) Heat maps showing association of CD8A correlated genes for 10 cell populations (Puram et al.) and 22 immune cell populations (Newman et al.). (e) Interferon gene signature in NPC cf. control, marked as orange, red, and blue for CD8+ low, CD8+ high NPC, and control, respectively. The p-values in (a-b) are depicted as = <0.01; = <0.005 * <0.001 ***= <0.0001. Abbreviations: NPC = nasopharyngeal cancer, GEO = gene expression omnibus.
This part was based on another data-set than that discussed by Reviewer 2.
This manuscript is a resubmission of an earlier submission. The following is a list of the peer review reports and author responses from that submission.
Round 1
Reviewer 1 Report
The present study describes a series of experiments designed to categorise NPC biopsies into different immune phenotypes and attribute these to overall survival. The study describes a thorough analysis of 48 biopsies, resulting in the characterisation of these biopsies as "inflamed", "excluded" or "deserted".
I was particularly interested in the difference between survival rates (DFS and DDS) between the inflamed and excluded phenotypes and how they correlated with EBV positivity. I think you could have drawn that out further in the discussion and perhaps commented on why this might be significant.
However, overall, I think this is an extremely interesting study defining immune phenotypes of the classic NPC lymphoepithelioma. The findings are robust, the data is presented well and there are only a few minor spell check/typos to address (please see comments in red on the attached manuscript).

Reviewer 2 Report
This study aims to investigate the immune phenotypes of nasopharyngeal cancer. The study suggests that NPC lesions are heterogeneous with regard to distribution of CD8+ T-cells and CD207+ DCs, it can be classified into immune phenotypes that carry prognostic information. In addition, CD207+ DCs may represent a target for immunotherapy with potential to facilitate the antigen cross-presentation necessary to execute cytotoxic T-lymphocyte responses. However, the relate article about CD8+ T-cells and on nasopharyngeal cancer have been published, although the association of CD207+ DCs with nasopharyngeal cancer still not clean. These findings described here is fewer interesting.
Major Comments
- All of 48 patients’ numbers were too few, it may be difficult to collect the quantity, but at least 200 is recommended.
- Suggest further provide the characteristics among study cohorts propensity score-matching populations.